# Dynamic Simulation and Modeling of a Novel NeuRaiSya for Railway Monitoring System Using Petri Nets

**DOI:** 10.3390/s24134095

**Published:** 2024-06-24

**Authors:** Bhai Nhuraisha I. Deplomo, Jocelyn F. Villaverde, Arnold C. Paglinawan

**Affiliations:** 1School of Graduate Studies, Mapua University, Manila 1002, Philippines; jfvillaverde@mapua.edu.ph (J.F.V.); acpaglinawan@mapua.edu.ph (A.C.P.); 2College of Computing and Information Sciences (CCIS), University of Makati, Makati 1215, Philippines; 3School of Electrical, Electronics and Computer Engineering, Mapua University, Manila 1002, Philippines

**Keywords:** Petri nets, reachability graph, railway, modeling and simulation, GreatSPN

## Abstract

This research introduces the NeuRaiSya (Neural Railway System Application), an innovative railway signaling system integrating deep learning for passenger analysis. The objectives of this research are to simulate the NeuRaiSya and evaluate its effectiveness using the GreatSPN tool (graphical editor for Petri nets). GreatSPN facilitates evaluations of system behavior, ensuring safety and efficiency. Five models were designed and simulated using the Petri nets model, including the Dynamics of Train Departure model, Train Operations with Passenger Counting model, Timestamp Data Collection model, Train Speed and Location model, and Train Related-Issues model. Through simulations and modeling using Petri nets, the study demonstrates the feasibility of the proposed NeuRaiSya system. The results highlight its potential in enhancing railway operations, ensuring passenger safety, and maintaining service quality amidst the evolving railway landscape in the Philippines.

## 1. Introduction

Railways are a climate-smart and efficient way to move people and freight deployed in most countries worldwide [1]. Railway monitoring systems encompass an extensive array of technologies, tools, and processes crafted for monitoring and overseeing diverse facets of railway operations. This system integrates a variety of sensors, communication devices, and software solutions, all geared toward guaranteeing the safety, efficiency, and dependability of railway networks. Railway operators globally are keen to enhance the efficiency and resilience of railway systems while prioritizing passenger safety and maintaining high-quality service. Technologies like automated train operations (ATO) and artificial intelligence (AI) stand prominently on the forefront of the railway innovation agenda in numerous countries [2]. In the Philippines, railway networks consist of the Philippine National Railway (PNR), Light Rail Transit Authority (LRTA), and Metro Rail Transit Corporation (MRTC), which are located in Luzon. The MRT first opened in 1999 and became fully operational in the year 2000. On the other hand, the LRT was elevated and built in January 2000. The Philippines National Railway (PNR) lines plus the LRT and MRT urban mass transit lines in Metro Manila make up the Greater Capital Region’s (GCR) present rail network. The PNR commuter service currently has 36 stations serving Metro Manila and Laguna. The LRT Line 1 has 20 operational stations extending from Roosevelt to Baclaran. Its extension project consists of 8 stations that will serve Parañaque down to Cavite. The LRT 2 has 13 stations extending from Recto to Antipolo. The extension will serve more of the Manila and Rizal areas. Lastly, the MRT 3 has 13 stations running in an orbital north-to-south route following the alignment of EDSA. The planned MRT 7 will serve the areas of Quezon City to Bulacan. The Philippine Railway Network of the Luzon and Mindanao Rail network has ongoing projects to build new transportation lines and upgrade the existing railway from Luzon to Mindanao such as North South Railway project’s (NSRP) South Line (Manila to Southern Luzon), Northrail project (Metro Manila to Central and Northern Luzon), LRT 1 North and South Extension project (Cavite Extension), MRT (Metro Rail Transit System) 3 Capacity Expansion, LRT 4 and 6 (Taytay Region to Manila), MRT 7 (Quezon to Caloocan), Mass Transit System Loop (Taguig, Makati, and Pasay), Mega Manila Subway (San Jose del Monte to the southern end of Dasmarias), Integrated Transport System (Metro Manila), and Mindanao Rail Network (Davao City) [3,4,5].

Figure 1 shows the conceptual framework of NeuRaisya System that simulated using the Petri Nets. The train movements can be monitored through the electronics modules that capable to send sensor values to the database. The number of passengers loading and unloading were captured by the camera installed to the train station with Neural Network application.

With the growing number of trains in the country, there is a corresponding increase in passengers, highlighting the importance of monitoring both trains and passengers. The researchers of this study developed the NeuRaiSya (Neural Railway System Application), a novel alternative railway signaling system with deep learning applications for passenger analysis. Neural networks are effective tools for modeling and solving complicated issues in a variety of disciplines. Deep learning is a subset of machine learning that focuses on neural networks with several layers, sometimes known as deep neural networks. One of the types of neural networks is a physics-informed neural network (PINN). It solves partial differential equations (PDE) with physical equations as operational constraints. The goal of a PINN is to translate physical limitations into extra loss functions in deep neural networks [6]. PINNs provide a new technique in simulating realistic physical flows in which some data are accessible from multimodality measurements, but the boundary or initial conditions are unknown. Using PINNs, there are numerous prospects for engineering research that can be used to make predictions like velocity, flow, pressure, and so on [7]. However, this paper is focused on the simulations and evaluations of the management or model process of the novel NeuRaiSya using the GreatSPN tool (Graphical Editor for Petri Nets). GreatSPN is a utility designed for the modeling and examination of Generalized Stochastic Petri Nets (GSPNs).

## 2. Petri Nets and Their Types and Applications

The Petri net (PN), conceived by Carl Adam Petri during the 1960s, serves as a valuable instrument for modeling and examining distributed systems. The Petri net has found applications in various scientific and technological domains, including computer science, automation technology, and mechanical design and manufacturing. The Petri net has four elements: place, transition, arc, and tokens.

**Definition** **1.**
*The PN described with this equation: PN = (P, T, I, O, m_0_). “P” denotes places (p ∈ P), representing status elements or conditions within the system and can be expressed as P = {p_1_, …, p_n_}. “T” signifies transition (t ∈ T) elements, representing occurrences, events, movements, signal exchanges, or actions that can occur in the system, and can be written as T = {t_1_,…,t_n_}. “I” stands for the input function from P (places) to T (transition). “O” denotes the output function from T to P. “M_0_” represents the initial markings, which is the initial distribution of tokens in places when the system starts [8,9].*


Petri nets can be categorized into three types: timed Petri nets, stochastic Petri nets, and colored Petri nets.

(1)Timed Petri nets (TPNs) integrate time as a crucial element, allowing the modeling of systems where the timing and duration of events and transitions are vital. This modeling approach includes the concept of time to represent systems where the duration of transitions and the timing of events are fundamental aspects. It includes time parameters associated with transitions, indicating the time required for transitions to occur. It is used for modeling real-time systems where the timing of events is critical. Object enhanced time Petri nets (OETPNs) extend beyond classical Petri nets (PNs) by incorporating tokens that model both passive and active objects (execution threads) [10].

Applications: Railway network utilization model (RNUM). This model evaluates the efficiency of a railway network, including factors such as the utilization of the network and performance metrics like scheduled wait times for a specific train line network & railway emergency plan modeling [11]. OETPNs are used for modeling a railway system with high-speed trains. This model is suited to design and analyze the resilience of railway systems [12].

**Definition** **2.**
*The transitions in TPNs can be represented using algebraic expressions like t ≥ MinTime and t ≤ MaxTime. Here, ‘t’ denotes the time elapsed since the transition became enabled, while MinTime and MaxTime signify the respective minimum and maximum time intervals linked to the transition.*


(2)Colored Petri nets (CPNs) extend the basic Petri net model by assigning colors or additional attributes to tokens, places, or transitions. This model uses different types or colors of tokens to represent different entities or conditions, providing a more expressive way to represent information. It offers flexibility in modeling systems with diverse and complex attributes, such as concurrent processes with different properties. CPNs reduce the complexity of modeling [13].

Application: Modeling and analysis of ETC control system. The researchers employed the CPN model for a more visually streamlined representation. This model not only allows dynamic simulation of business processes but also captures real-time information generated by the system. In comparison to alternative modeling tools, it offers a more straightforward observation of its control process, facilitating the application of formal methods to analyze the correctness of the system model [14].

**Definition** **3.**
*In CPNs, algebraic expressions based on tuples are used to define conditions and actions associated with transitions. It has a nine tuple CPN = (P, T, A, Σ, V, C, G, E, I), where [11,12]:*
*(a)* 
*P is a finite set of places.*
*(b)* 
*T is a finite set of transitions such that P ∩ T = ϕ.*
*(c)* 
*A ⊆ (P × T) ∪ (T × P) is a set of arcs from place to transition and from transition to place, which indicates where the token flows.*
*(d)* 
*Σ is a set of color sets. This set contains all possible colors, operations, and functions used within the colored Petri net.*
*(e)* 
*V is the finite set of the type of variables.*
*(f)* 
*C: P → Σ is a color function. It maps places in P into colors in Σ.*
*(g)* 
*G: T → Λv is a guard function that assigns a guard to each transition.*
*(h)* 
*E: A → Λv is an arc expression function. It maps each arc a ∈ A into the expression e.*
*(i)* 
*I is an initialization function. It maps each place p into an initialization expression i.*



(3)Stochastic Petri nets (SPNs) introduce random or probabilistic elements to model systems with uncertainty or variability in the occurrence of events. This model assigns probabilities to transitions, indicating the likelihood of their occurrence. It is useful for modeling systems where events have probabilistic characteristics, such as communication networks or manufacturing systems. The extension of SPNs is Generalized Stochastic Petri Nets (GSPNs). The GSPN model is the conventional Generalised Stochastic Petri Net formalism for describing the operational behavior of a dynamic system. It is a graphical and mathematical framework designed for the modeling of concurrent and distributed systems.

Application: SPNs have been used in several application areas including fault diagnosis, maintenance logistics planning and optimization, and reliability and availability analysis of the railway [15,16,17,18].

**Definition** **4.**
*The GSPN is described with a 6-tuple (P, T, F, W, Mo, Λ), where*

*P = {p*
_1_
*, p*
_2_
*, … , p_n_} is a finite set of places; T = {t1, t2, … , tn} is a finite set of transitions; F ⊆ (P × T) ∪ (T × P) is a set of arcs; W: F → {1, 2, 3, …} is a weight function; M_0_: P → {0, 1, 2, 3, …} is the initial marking; Λ = {λ_1_, λ_2_, … , λm} is the set of firing rates associated with the transitions; P ∩ T = ϕ and P ∪ T 6 = ϕ [19].*


### Different Simulation Tools of Petri Nets

Train simulations employ the motion model to replicate the dynamics of a train in various real and hypothetical situations. For example, they are applied in capacity evaluations for current and prospective railway lines, determining track section occupancy and arrival times for input into timetabling tools. Simulators are also utilized for estimating energy consumption, aiding in infrastructure planning, testing new signaling systems, and providing training for drivers [20].

(a)TINA simulation is a tool designed for the examination of Petri nets and timed Petri nets. Petri nets are used as a formalization tool to model different train situations, such as overtaking, following, and station avoidance. The reachability graph of the Petri net model is calculated using the TINA version 3.8.0 simulation software, enabling the determination of whether the given train operation scheduling meets station requirements [21].(b)“PetriNet Editor + PetriNet Engine” is based on the open-source Petri net editor PNEditor 0.9.2, which has been created for the support of modeling of systems using timed interpreted Petri nets. As a result, it enables the implementation of control algorithms on Arduino-type microcontrollers and other compatible microcontrollers as well [22].(c)PNet is used an alternative pure Python 3.0 library for Petri net modeling by reducing its object-oriented programming overheads to its minimum and adding Python functions as an alternative type of transition rule. Hence, PNet is expected to make it easier for beginners to start working with Petri nets before moving on to more comprehensive libraries like SNAKES. PNet has been incorporated into COPADS, a library of algorithms and data structures, developed entirely in the Python programming language and has no third party [23].(d)PetriBaR is a MATLAB toolbox version 1.0.0.0 (12.4 KB) used for the analysis and control of Petri nets. PetriBaR is a collection of functions designed for fundamental Petri net analysis, including computing T-invariants, siphons, and reachability graphs, as well as performing monitor design, reachability analysis, state estimation, fault diagnosis, and opacity verification [24].(e)ORIS software 1.0 presents an exclusive approach for quantitatively modeling and analyzing non-Markovian models. It incorporates a novel graphical editor and a Java library that facilitates both transient and steady-state analysis. The software architecture, tailored to integrate new features of Petri models and advanced analysis methods, positions ORIS as a versatile research tool for assessing innovative solutions in discrete-event systems. ORIS has demonstrated successful applications across diverse contexts and application domains, serving, for example, as a graphical user interface for evaluating performability measures in railway signaling systems [25].(f)CPN Tools 4.0 (Colored Petri Net Tools) is a software bundle designed for the creation, simulation, and examination of colored Petri nets. Featuring a user-friendly interface, it facilitates both qualitative and quantitative analyses. It is used for assessing the performance of wireless sensor networks. CPN Tools was utilized to scrutinize quantitative properties through the monitor technique [26,27].(g)PIPE2 4.3.0 (Platform Independent Petri Net Editor 2) is a modeling instrument for Petri nets, enabling users to generate, modify, and scrutinize Petri net models. It accommodates various types of Petri nets. PIPE2 was used to design a Petri net model of Unistar CSV24 [28].(h)GreatSPN 3.1 (Graphical Editor for Petri Nets) is a utility designed for the modeling and examination of Generalized Stochastic Petri Nets (GSPNs). It facilitates both qualitative and quantitative evaluations of system behavior. It is used for performance evaluation by predicting performance parameters. GreatSPN emerged in the 2000s as a tool for constructing Petri nets, and over time, it has undergone significant development, incorporating various features. It has evolved to support formalisms like stochastic, colored, and stochastic colored, as well as the commonly used simple place and transition formalism. GreatSPN provides a range of tools to simplify the creation of system representations, including features for generating reachability graphs, verifying for dead markings, and facilitating the token [29,30].

In this study, the researchers utilized GreatSPN for simulation and modeling the NeuRaiSya system. GSPNs could incorporate nonlinear blocking effects that were not accounted for in previous analytic, probabilistic, and queuing-based approaches. To validate the model’s efficacy, simulations were performed within the context of a medium-sized railway station. The Generalized Stoichiometric Petri Net model was used for the railway station, validated by comparing its closeness to real-world data [31].

Figure 1 shows the conceptual framework of the study. The inputs are the passengers and the train. Modules were deployed in the cabin and railway station to monitor the position and the status of the train. The number of passengers loading and unloading were captured by the camera installed in the station. The parameter values from the wireless sensors were received by the transceiver gateway and then sent to the station module with a database. The captured videos were fed to the neural network, and the deep learning architecture was used to monitor and predict the number of passengers in the stations. The results were displayed to the GUI (graphical user interface) of the developed signaling system. The integration of deep learning techniques in passenger analysis within railway systems represents a pivotal advancement in optimizing transportation services. In the context of railway passenger analysis, deep learning enables the extraction of valuable insights from complex datasets, encompassing diverse facets of passenger behavior and movement patterns. In the NeuRaiSya, a selection of deep learning algorithms tailored to the specific requirements of the task is paramount. Once the number of passengers are recognized and classified, the number of passengers are fed to the network for the prediction process using a deep convolution neural network (DCNN). Convolutional neural networks (CNNs) would be ideally suited for analyzing closed-circuit television (CCTV) footage to identify passenger activities and crowd dynamics within train stations. The NeuRaiSya draws from diverse data sources, including CCTV footage and passenger counting sensors. Interpreting the outputs of trained deep learning models would show intricate passenger behavior patterns, crucial for informed decision-making in railway management. The passengers’ videos captured from the database are tested and pre-trained by 80/20: the process of the learning model; (1) the pre-processing; (2) data analysis; and (3) deep learning analysis. The validation method used is the confusion matrix by using variables such as precision, recall, and accuracy. The following equations give a description of them, with TP, TN, FN, and FP standing for true positive, true negative, false negative, and false positive, respectively.
$$precision=\frac{TP}{TP+FP}$$
$$recall=\frac{TP}{TP+FN}$$
$$accuracy=\frac{TP+TN}{TP+FP+TN+FN}$$

## 3. NeuRaiSya System Model

The research proposes a method to monitor the number of passengers within the station at a given time, monitor the speed and location of the train, and signal a halt whenever the system encounters a train issue.

Figure 2 illustrates the steps for simulating and modeling the NeuRaiSya system using Petri nets from step 1 to step 9, such as: (1) system design; (2) setting simulation parameters; (3) simulation and modeling of different scenarios; (4) definition of Dynamics of Train Departure model; (5) definition of Dynamics of Train Operations with Passenger Counting model; (6) definition of Timestamp Data Collection model; (7) definition of Train Speed and Location model; (8) definition of Train Related-Issues model; and (9) result analysis. The initial stage is system design, involving the creation of the overall system design and outlining its structure, components, and interactions. The second stage entails setting simulation parameters, which involves identifying system parameters such as passengers, train speed, train location, and sensor data. Following this, the simulation and modeling of different scenarios involve the use of simulation techniques to model diverse scenarios, exploring various situations and conditions within the designed system. Subsequently, the definition of the Dynamics of Train Departure model involves developing a Petri net model focused on capturing and defining the dynamics associated with the departure phase of the train. The subsequent step is the definition of the Train Operations with Passenger Counting model, aiming to construct a model, potentially utilizing Petri nets, to define and simulate the dynamics of train operations, with an emphasis on aspects related to passenger counting. The definition of the Timestamp Data Collection model is the subsequent step, involving the establishment of a model for collecting timestamp data, and representing and simulating the time-related aspects of the system. Further steps include formulating a model that defines the dynamics of train speed and location, creating a model specifically dedicated to defining and simulating issues related to train operations using Petri nets to represent problem scenarios. Lastly, the analysis of outcomes and results obtained from the simulations involves evaluating the performance, efficiency, and potential issues within the system based on the defined models and scenarios.

### 3.1. Petri Nets Model of the Study Based on the Train with Related Issues

In Petre nets, it is important to identify the places (P) to represent states of the system and transitions (T) to represent events or activities and to indicate the flow of tokens, representing the occurrence of events or state changes, as tabulated in Table 1.

Figure 3 shows the Petri net model of the study of the combination of the following models: (1) Train Departure; (2) Train Operations with Passenger Counting; (3) Timestamp Data Collection; (4) Train Speed and Location; and (5) Train Related-Issues. This study has identified places (P) from P_1_ to P_20_ and transitions (T) from T_1_ to T_20_.

The initial token is located at P_15_, which indicates that the train is ready to enter the station. Another token is located at P_12_ and P_6_, indicating that the recording of the leaving train was successful and the signal for train acceptance is ready.Upon firing T_1_, this specifies that the train is moving from one station to another. The tokens would move from their initial position to P_2_, P_7_, and P_9_. The following places indicate that the station is in the process of accepting a train, enabling human counting via a CCTV monitoring system, and that its arrival time is successfully recorded.Firing T_2_ indicates that the train would stop at the station to unload passengers. The token then moves to P_3_ and P_5_, opting for the train doors to open and the train to be stationary.For the train to reach its state for moving out of the station (T_4_), T_3_ and T_5_ must be fired, which specify that the train could unload its passengers and that the data for the train arrival were successfully sent to the database.Firing T_4_ moves the token to P_6_, P_11_, P_8_, and P_13_. P_6_ implies that the station is ready to accept another train, while P_11_ describes the train’s leave time recording. Subsequently, P_13_ shows that the sensors connected to the train are ready to record the train’s discrete data. P_8_, on the other hand, indicates that the train may travel to the next station or undergo repairs if necessary.If the train undergoes regular operation and encounters no technical problems, transition 8 should be fired. For T_8_ to be fired, the train’s velocity, acceleration, speed, and location should be recorded. A token at P_14_ would indicate the success of the process.If the train experiences technical issues, the train token should be moved to P_16_ to prompt troubleshooting of the train.While at this position, transitions T_12_ and T_13_ can be fired. Minor repair stations constitute T_12_, while major repairs constitute T_13_. Firing T_12_ instigates a resolution for minor repairs. This would lead to the train going back to regular operation.If significant repairs are required, T_13_ is to be fired. The train token moves to P_18_, which stipulates that the maintenance crew is doing assistance. Simultaneously, data are delivered indicating that the train is experiencing an issue, as described by P_19_. During this process, the station cannot accept another train since significant repairs are being performed (described in the Petri net system by an inhibitor arc).The tokens then move to P_20_, indicating that the train is receiving major repairs and is ready for final checking (T_15_). The train would undergo normal operations after the major repairs, prompting the whole system to repeat its processes.

The additional system is the troubleshoot (P_16_) system, which is comprehensive enough to handle issues from multiple stations; it also requires connections. As seen in the model, tokens move along a path, indicating the flow of processes from one station to another. Each station’s process, from signal to troubleshooting, is similar, with variations depending on issues encountered. The troubleshoot (P_16_) system effectively manages these issues, directing maintenance efforts and returning trains to operational status. The Train with Related-Issues model has two transitions that would determine whether the issue encountered is major or minor. Handling minor issues is described using a linear graph. Handling major issues, on the other hand, incorporates a place connected to T_4_ using an inhibitor arc. This indicates that the station would not allow another train to fire to the station where the central issue is situated. For the transition T_4_, which is connected via an inhibitor arc, an equation describes it as:$$t\;is\;enabled\;\forall I\left(P_{n},T_{n}\right)\backslash \left(P_{ni},T_{ni}\right),M\left(Pn\right)=1$$

Therefore,
$$Enabled\left(t\right)=\prod_{n=1}^{I(P_{ni})}I\left(P_{ni},Tn\right)=0$$

This equation indicates that a place connected to a transition by an inhibitor arc should contain zero tokens for that transition to be fireable. This system represents the function of a repair facility, wherein a train cannot enter a station if a train experiences a major issue.

### 3.2. Dynamics of Train Departure

Figure 4 shows the Petri nets of the Train Departure model. The token located at P8 indicates the train is moving out of a station. This train can take two paths, one for the station and the other for the parking system. For the transition T1 to work, another token located at P6 denotes a signal that a train is entering the station. The train would simultaneously go stationary (indicated by P5) and allow the unloading of passengers (P3 to P4). The train is now ready to move out of the station again. For the transitions to be allowed to fire, it would require prior places to have one token described by the equation:
Tn is t if I(Pn,Tn)=1
$$\mathrm{or}\;\mathrm{Enabled}(t)=\prod_{n=1}^{I(Pn)}M\left(Pn\right)=1$$
where:

*M* be the marking of tokens

*P* be the place

*T* be the transitions

*t* be the firing of tokens

*I* be the input

*Tn* = number of transition number

This equation indicates that a transition (*Tn*) is fireable if and only if the input (*I*) from a place (*Pn*) to a transition has a token value of one. This equation will hold true for all transitions within the system. If a transition has connections from multiple places, it would only fire if each place has one token within it. For the Train Departure model, P8 and P6 contain a token, which indicates the train and data (for enabling train arrival), respectively.

### 3.3. Dynamics of Train Operations with Passenger Counting Model

Petri Net models should include a way to monitor the humans within the station. The CCTVs within the station capture live data and send them to a database. This can be implemented in the database in this way. The data could then be used for predicting the number of people that will be at the station.

Figure 5 shows the Train Operations with Passenger Counting model in train operation. CCTV cameras allow the system to count the number of passengers entering the station. P_7_ denotes human counting, which would operate simultaneously as the train enters the station. When the train returns, P_8_ (MovingOut) and P_6_ (Signal) are still the ones with the initial tokens; they still function the same way, the P_6_ (Signal) just means that a train is not occupying the station. It then gives a token to HumanCounting (P_7_), which means that it is now counting the number of people within the station. Once the T_1_ (Ready) transition fires, it gives a token to P_2_ (MovingIn), indicating that the train is in a moving state. Finally, it fires T_4_ (ReadyOut) and sends a piece of token to P_6_ (Signal) and P_8_ (MovingOut). And again, we arrive at our original or initial position

### 3.4. Timestamp Data Collection Model

Figure 6 shows the Petri nets for the Timestamp Data Collection model. The system incorporates a timestamp collection model that would allow the gathering of data for the arrival and leaving time of the train within the station. The recording for train arrival would operate as the train enters the station. The function is described by the linear process from P_9_ to P_10_. The data for the leaving time of the train from the station would operate linearly from P_11_ to P_12_. A token is introduced in this model located at P_12_, which implies the data for recording the train departure. Sensors data are sent and recorded successfully to the database once the train has departed.

### 3.5. Train Speed and Location Model

Figure 7 shows the Petri nets of the Train Speed and Location model. The additional system created to record the train’s speed and location is an Arduino component/sensor embedded within the train that would send data about the location, speed, acceleration, and other necessary data. The operation would protrude as the train travels along the tracks.

## 4. Results and Discussion

The study would utilize reachability analysis to determine the train station system function. Reachability analysis describes the function of each station system and determines its plausibility when incorporated as a whole. GreatSPN was utilized in this study to create the reachability tree for each system. The figures indicate that the suggested system has no dead markings, signifying that the system is reachable/plausible to operate.

The reachability tests were tabulated in Table 2 per module, such as: (a) Train Departure; (b) Train Operations with Passenger Counting; (c) Timestamp Data Collection Model; (d) Train Speed and Location; and (e) Train Related-Issues Model

Each model was tested using reachability tests for both GreatSPN simulation and manual computing. Figure 8, Figure 9, Figure 10, Figure 11 and Figure 12 are shown below.

Figure 8a shows the reachability test results using GreatSPN simulation, while Figure 8b shows the calculated result of incidence matrix of the Train Departure model. The topmost row provides the legend for the transition, while the leftmost column indicates the legend for the places present within the model. The highlighted column on the rightmost part of the table gives the system’s initial marking, located in P_8_ and P_6_. This matrix shows transition markings compared to place markings. The positive value (1) implies that a place would input a token to a transition. A negative value (−1) means that a token would be extracted from a transition to a place. The incidence matrix can be utilized to determine whether the termination of a process is present. It requires the marking at any point of the model to be multiplied to the converted incidence matrix to determine whether termination is present. The equation for checking whether a transition can be terminated is given by
$$E=M*I_{T}$$
wherein *E* indicates the vector with the enabled transition. *M* is the current marking or any marking to be checked. *I_T_* is the incidence matrix. It is essential to transpose the incidence matrix to a suitable form in case the matrix multiplication is not enabled. The resultant vector shows no instance wherein all vectors are zeros, indicating that the system does not reach termination. If no termination is achieved, the system is reachable. It indicates the path each token takes if the firing rule is followed. The line transitions suggest that the model is reachable since it shows that initial tokens located at P6 and P8 would return to their original positions. GreatSPN software can generate an automatic version of the reachability tree, and it can automatically show if there are terminations for a system. A termination implies that there is no fireable transition left within a system or *Tn* is *t* if = 0. 3.

Figure 9a illustrates the reachability test result of Model 2, while Figure 9b presents its incidence matrix pertaining to the Train Operations with Passenger Counting model. The figures also show that the proposed system is fireable due to it not having an instance wherein it would reach termination, or no transitions could fire due to places not having a token. It is essential to indicate that an additional token would protrude from transition 2.

Figure 10a,b shows that the proposed addition of the Timestamp Data Collection model from the base train is also reachable, indicating the proposed system’s plausibility. The reachability tree shows multiple branches protruding from transition 1, where the timestamp data starts its function. GreatSPN can determine whether the system would have an instance of termination. The addition of a timestamp did not affect the reachability of the system as a whole. Figure 10b shows the incidence matrix for the Timestamp Data Collection model for the train station. An additional transition and place were added within the matrix to represent the additional system for the Data Collection model. It is essential to point out that an extra token was added to P_12_ to show the data for recording train departures, hence added in the initial marking Mo.

The additional Train Speed and Location model, as shown in Figure 11a, did not affect the feasibility of the proposed system. It can be observed that the integrated system protrudes from transition 4, generating a token that indicates the data for recording the location, speed, acceleration, and other necessary data. This system introduced additional transitions and places, making the overall matrix system larger. Positive numbers show the token entering a transition, while negative numbers indicate a token leaving a transition. The initial marking also changed in this matrix, wherein a token was introduced in P_1_ to show the data for recording parameters for location, speed, acceleration, and tilt angle.

The reachability tree, as shown in Figure 12a, with the addition of a train-related system within the base train station, indicates a similar outcome to the previous model. GreatSPN software suggests that the additional system proves to be reachable since there are no dead ends, which would lead to the termination of the whole process. The software is having problems with creating the entire reachability tree; however, the results show that there would still be no termination since the graph panel does not show a dead end within the tree. The final incidence matrix, as shown in Figure 12b, is for the overall system of the train station Petri net system model with the Train Related-Issues model. It shows the overall movement of the token for each system introduced within the base model. It can be observed that the movement for each token within sets of transitions is positive when a token goes into a transition and negative when it goes out of a transition and into a place.

Modeling and simulating systems with Petri nets holds significant importance for several reasons. Firstly, these nets offer both visual and mathematical representations of intricate systems, which enhances comprehension of their behaviors and interactions. This aids in conceptualizing, designing, and evaluating systems prior to their actual implementation. Secondly, Petri nets facilitate simulation, allowing researchers and engineers to forecast system behaviors under varied conditions without resorting to costly or hazardous real-world experiments. This capability expedites testing and optimization processes, resulting in systems that are more efficient and dependable. Moreover, Petri nets are instrumental in pinpointing bottlenecks, deadlocks, and other inefficiencies within systems, thereby streamlining processes and enhancing overall performance.

Conducting a comparative analysis between the NeuRaiSya and existing railway monitoring and signaling systems stands as a critical step in comprehending its advantages and potential impact. The innovative essence of the NeuRaiSya resides in its integration of deep learning algorithms and advanced sensor technology to procure real-time data on train movement and passenger activity. Unlike conventional systems, which rely on rudimentary methodologies like manual monitoring or basic sensor technology, the NeuRaiSya presents a more exhaustive and precise approach to railway management. A notable advantage of the NeuRaiSya lies in its economically feasible implementation. Whereas extant systems may necessitate considerable financial outlay for installation and upkeep, the NeuRaiSya offers a cost-effective alternative, particularly advantageous for railway networks in developing nations like the Philippines. This affordability creates avenues for smaller rail operators to embrace sophisticated monitoring and signaling systems, thereby ameliorating safety and efficiency across the industry. Furthermore, the NeuRaiSya’s utilization of deep learning facilitates predictive analytics, enabling proactive decision-making in train scheduling and passenger management. In terms of scalability and adaptability, the NeuRaiSya presents flexibility to accommodate diverse railway infrastructures and operational requisites. Its modular design facilitates seamless integration with extant systems, mitigating disruptions during implementation and ensuring compatibility with forthcoming upgrades or expansions.

The research on formulizing a train station model using the Petri net system focuses on integrating multiple systems into a base station model and assessing their functions when introduced to actual scenarios. The researchers utilized reachability analysis and a reachability tree to determine the possibility of the actual process. The peripheral systems, such as the Train Operations with Passenger Counting, Timestamp Data Collection, Train Speed and Location, and Train Related-Issues models, are each integrated within the base train station model to evaluate the possibility of each process. The simulation results via GreatSPN indicate that the base station is reachable due to it having no deadlock states. Similar results are observed when integrated with peripheral systems. The reachability tree, although more complex when each system is integrated, shows to be reachable due to its lack of deadlock areas. The firing equations can also be used to assess the fireability of each transition, and similar results are shown. Overall, the proposed train station model with integrated peripherals using Petri net and reachability analysis is feasible and can be utilized as a basis when creating an actual train station due to its formulization being reachable via GreatSPN. Therefore, this study met the objectives. The simulations and modeling using Petri nets of each scenario emphasize that the process design proposed is achievable. The results of simulation are matched with the computed manual method using the incidence matrix formula.

Acknowledging the limitations of the current study and outlining potential areas for future research is essential for providing a balanced view and guiding the development of the NeuRaiSya. Firstly, discussing the practical challenges and limitations encountered during the implementation of the NeuRaiSya in real-world railway networks, such as data collection difficulties, sensor reliability, interoperability with existing systems, or regulatory barriers, is crucial. It should be acknowledged that while the study demonstrates the feasibility of the system, further refinement and adaptation may be necessary for widespread deployment. Addressing ethical considerations and data privacy issues is crucial in deploying technologies like deep learning for passenger analysis in public systems. The handling of passenger data within the NeuRaiSya prioritize integrity and confidentiality throughout the entire process. The researchers of this study coordinated with railway managers and operations about the data privacy. No data information from the passengers were gathered by the researchers during the testing of the system.

## Figures and Tables

**Figure 1 sensors-24-04095-f001:**
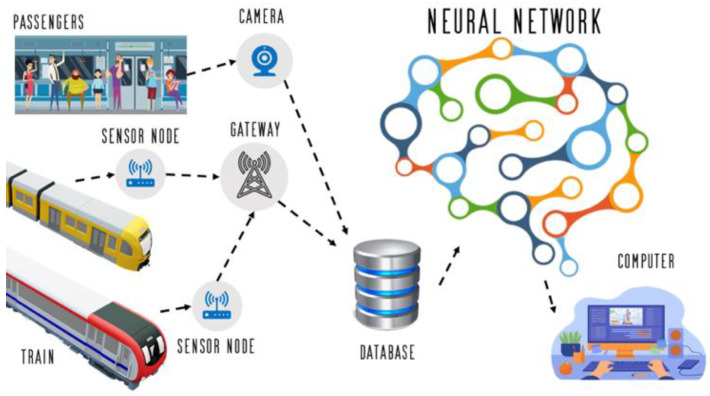
Conceptual framework of the NeuRaiSya System.

**Figure 2 sensors-24-04095-f002:**
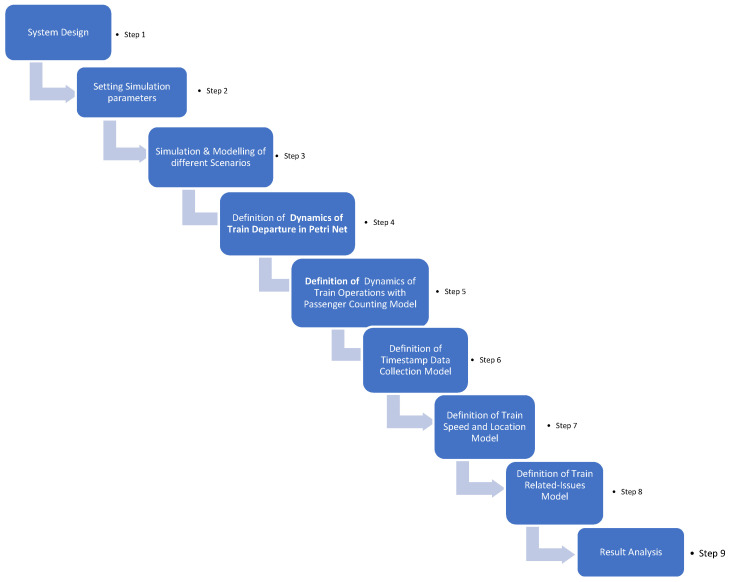
Methods of Simulation and Modeling of NeuRaiSya system.

**Figure 3 sensors-24-04095-f003:**
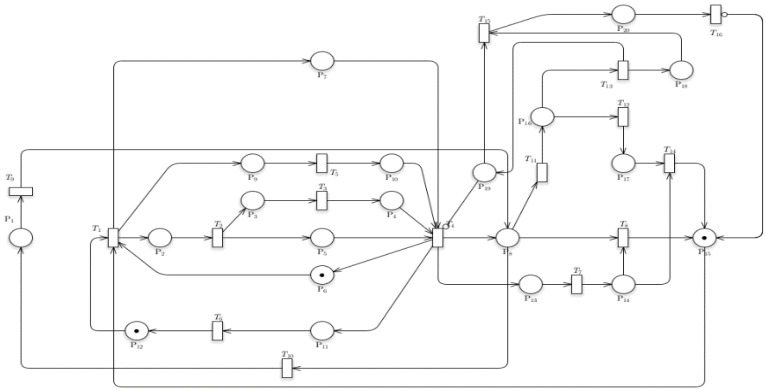
Petri nets of the Model 5 -Train with Related-Issues.

**Figure 4 sensors-24-04095-f004:**
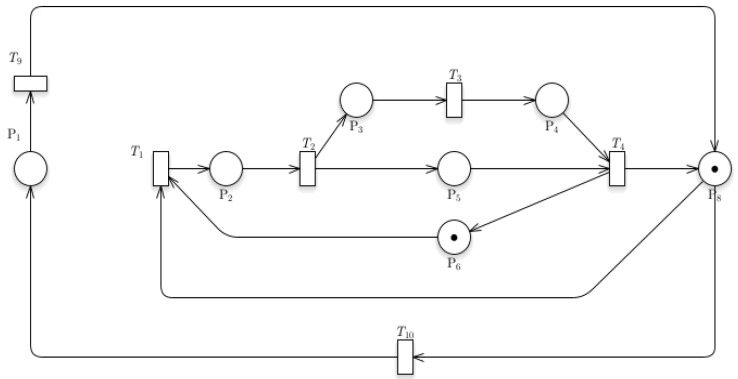
Petri nets of Train Departure model.

**Figure 5 sensors-24-04095-f005:**
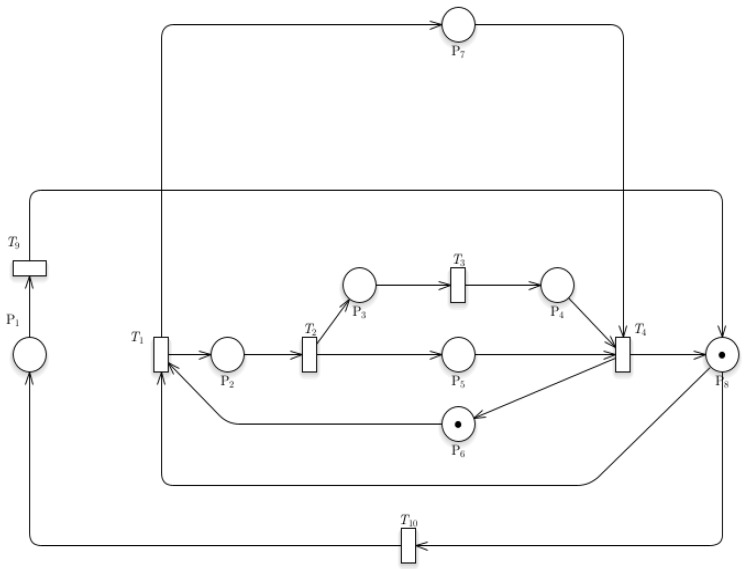
Petri nets of Train Operations with Passenger Counting model.

**Figure 6 sensors-24-04095-f006:**
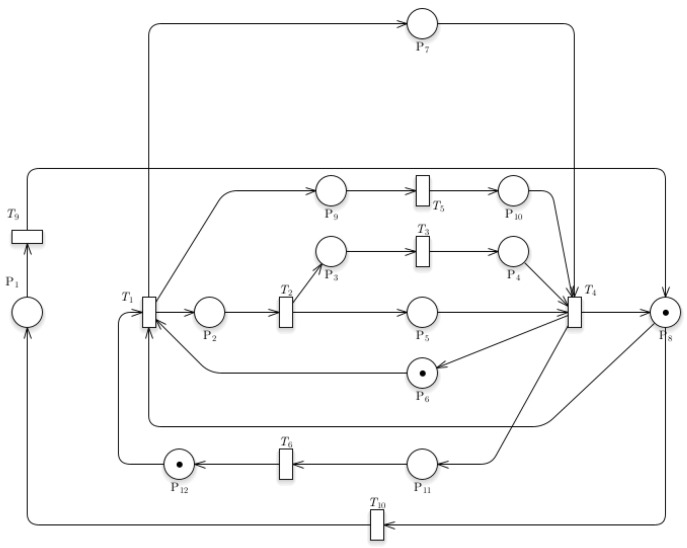
Timestamp Data Collection Petri nets.

**Figure 7 sensors-24-04095-f007:**
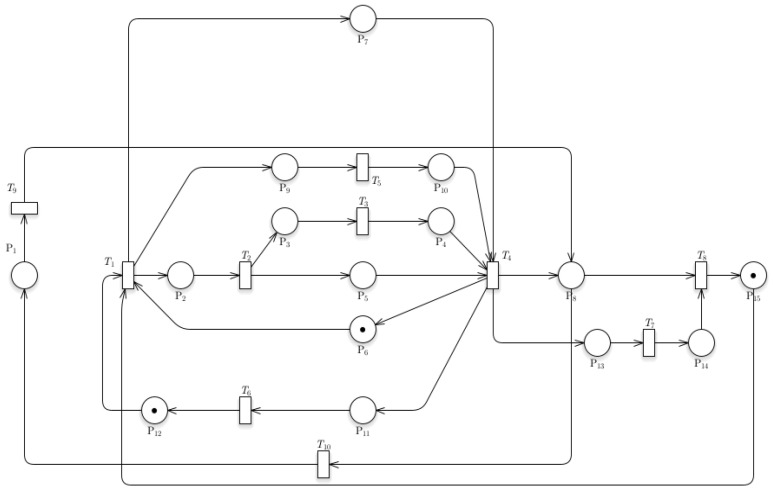
Train Speed and Location model Petri nets.

**Figure 8 sensors-24-04095-f008:**
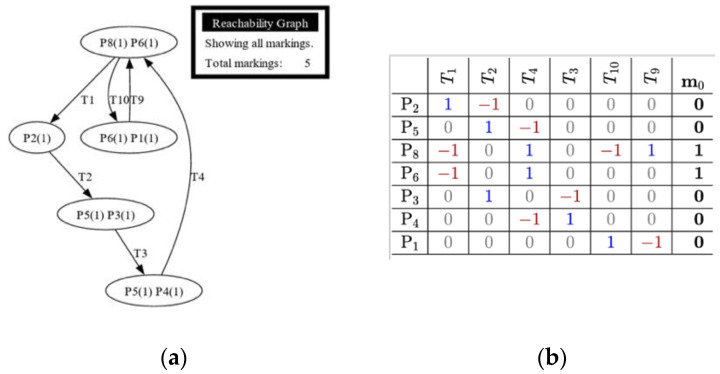
(**a**) Reachability graph of Model 1; (**b**) incidence matrix of Model 1.

**Figure 9 sensors-24-04095-f009:**
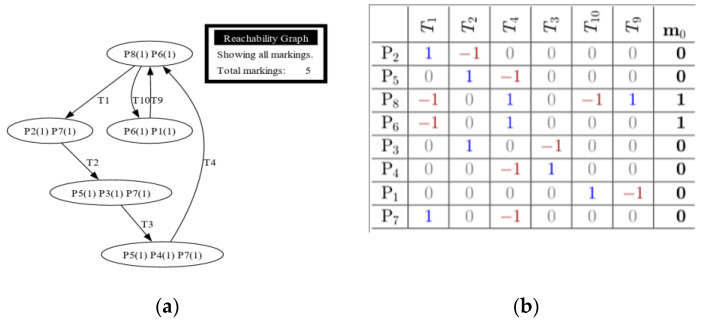
(**a**) Reachability graph of Model 2; (**b**) incidence matrix of Model 2.

**Figure 10 sensors-24-04095-f010:**
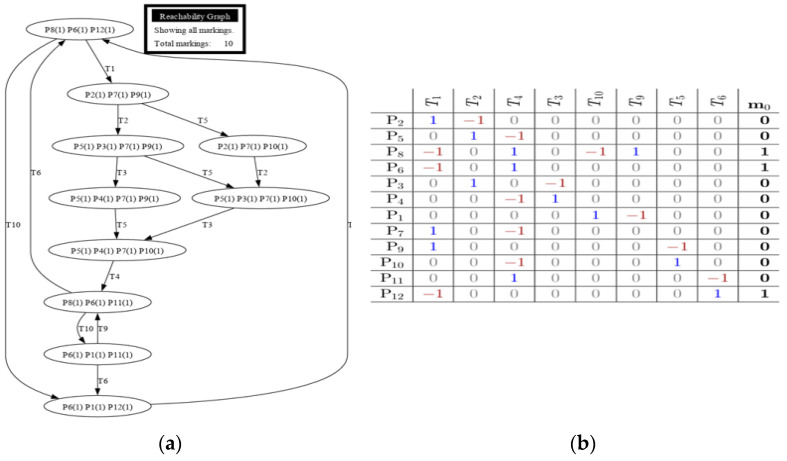
(**a**) Reachability graph of Model 3; (**b**) incidence matrix of Model 3.

**Figure 11 sensors-24-04095-f011:**
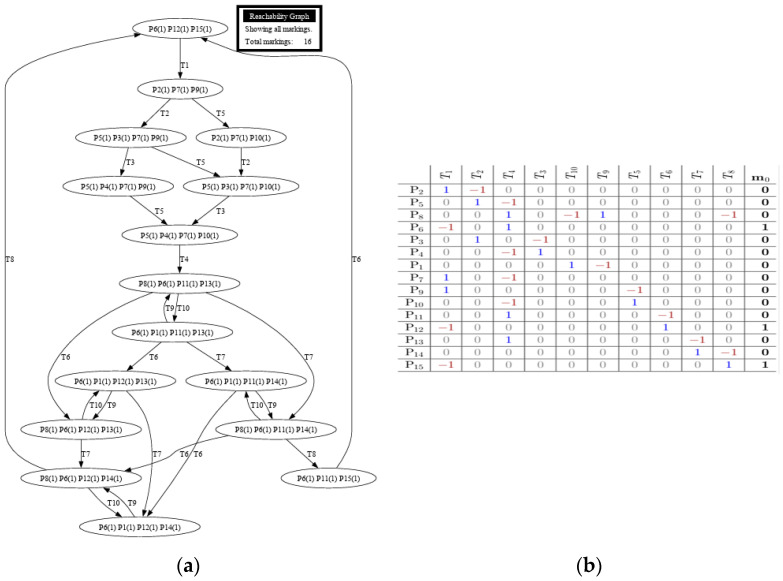
(**a**) Reachability graph of Model 4; (**b**) incidence matrix of Model 4.

**Figure 12 sensors-24-04095-f012:**
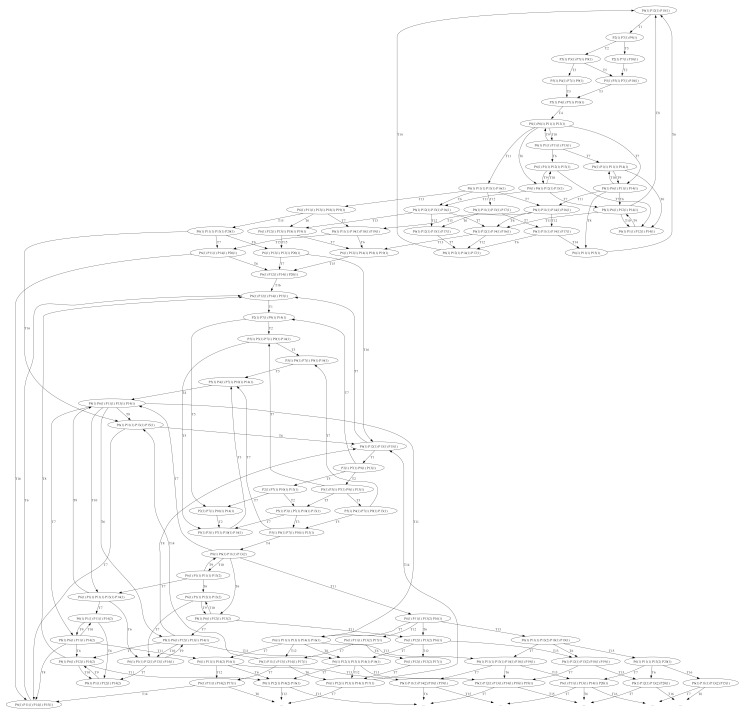
(**a**) Reachability graph of Model 5; (**b**) incidence matrix of Model 5.

**Table 1 sensors-24-04095-t001:** The places and transition of the system.

Place Name	P/T	Meaning
Park	P_1_	The train is parked/waiting for the signal from the station/train ahead if it is occupied.
MovingIn	P_2_	While this is enabled, a station is currently letting in a train. The specific station is dependent on where it is located.
Unloading	P_3_	While this is enabled, the station is letting the train unload its passengers.
Loading	P_4_	While this is enabled, the station is letting the train load its passengers.
Stationary	P_5_	While this is enabled, the train within that station is not moving.
Signal	P_6_	While this is enabled, the station is ready to accept new trains within itself.
HumanCounting	P_7_	While this is enabled, the station is recording the amount of people within itself.
MovingOut	P_8_	While this is enabled, the train is mobile and can have two possibilities during travel issue or no issues.
RecordArrivalTime	P_9_	While this is enabled, the station is recording the time the train has arrived.
ArrivalRecorded	P_10_	While this is enabled, the train’s arrival has been successfully recorded.
RecordLeave-Time	P_11_	While this is enabled, the station has recorded the last train’s leave time and is thus ready to accept the next train.
LeaveRecorded	P_12_	While this is enabled, the train’s departure has been successfully recorded.
RecordArdData	P_13_	While this is enabled, the Arduino within the train is reading the speed, location, angular velocity, and acceleration of the train it is on.
ArdDataSuccess	P_14_	While this is enabled, the Arduino data have been successfully recorded.
Arrival	P_15_	While this is enabled, the train is ready to enter the station.
Troubleshoot	P_16_	While this is enabled, the system is currently troubleshooting the train’s issues with the help of the crew.
Resolved	P_17_	While this is enabled, the train has resolved the issues it has faced.
Assistance	P_18_	While this is enabled, the train is receiving help to get itself out of the railway to receive maintenance.
ExperiencingIssue	P_19_	While this is enabled, the train has encountered an issue and is thus trying to troubleshoot itself.
Maintained	P_20_	While this is enabled, the train has finished receiving maintenance.
Ready	T_1_	While this is enabled, the train is ready to move.
Stop	T_2_	The train goes from moving train to stop status.
Wait	T_3_	The train unloads the passengers.
Ready or Ready Out	T_4_	After the train loads the passengers, the train is ready to move.
SendArrivalToDatabase	T_5_	Send the sensors data to the database once the train has arrived.
SendLeaveToDatabase	T_6_	Send the sensors data to the database once the train has departed.
SendArdToDatabase	T_7_	Once the Arduino data have been recorded, they are sent to the database.
JourneyEnd	T_8_	This transition fires when the train’s journey has been safe and has ended with no issues.
Send to Parking	T_9_	Send the train to the parking area.
Send out to Parking	T_10_	Send the train out of the parking area.
Send to Repair Station	T_11_	Send the train to the repair station for troubleshooting.
Send to Minor Repair Station	T_12_	Send the train to the minor repair station if the issue is non-fatal.
Send to Major Repair Station	T_13_	Send the train to the major repair station if the issue is fatal.
Send out of Repair Station	T_14_	Send the train out of the repair station if the issue is addressed.
Send to Final Checking	T_15_	Send the train to final checking if assistance to major issue is addressed.
JourneyEnd	T_16_	This transition fires when the train’s journey has been safe and has ended with no issues.

**Table 2 sensors-24-04095-t002:** Model 1 to Model 5.

Model Number	Model Scenario	Reachability Status
Model 1 Model 2	Train Departure Train Operations with Passenger Counting	Reachable Reachable
Model 3 Model 4 Model 5	Timestamp Data Collection Train Speed and Location Train Related-Issues	Reachable Reachable Reachable

## Data Availability

The data presented in this study are available on request from the corresponding author.

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
