# Peer review of "Dynamic Simulation and Modeling of a Novel NeuRaiSya for Railway Monitoring System Using Petri Nets"

_sensors, 2024, doi:10.3390/s24134095_

Round 1
Reviewer 1 Report
Comments and Suggestions for Authors
The paper presents a novel railway signaling system named NeuRaiSya, which integrates deep learning for passenger analysis to enhance the safety and efficiency of railway operations in the Philippines. The authors use the GreatSPN tool to simulate and evaluate the system's effectiveness through Petri Nets modeling. The study demonstrates the feasibility of NeuRaiSya through the design and simulation of four scenario models: Train Departure, Train Operations with Passenger Counting, Timestamp Data Collection, Train Speed and Location, and Train Related-Issues. I have the following comments and questions regarding this paper:
1. The paper could benefit from a more detailed explanation of the methodology, particularly the deep learning techniques used for passenger analysis. It would be helpful to include information on the algorithms, data sources, and validation methods to strengthen the credibility of the proposed system.
2. Currently, physics-informed neural networks have been widely applied in many engineering fields, as evidenced by recent publications such as "Engineering Structures, 2023, 297: 117027", "Computational Mechanics, 2021, 67: 207-230.", "Reliability Engineering & System Safety, 246, 110081." These latest advancements should also be adequately highlighted in the introduction.
3. A comparison with existing railway monitoring and signaling systems would provide a clearer picture of the advantages and potential impact of NeuRaiSya. This could include a discussion on how NeuRaiSya addresses limitations or challenges present in current systems.
4. The paper should address the scalability of NeuRaiSya to larger and more complex railway networks beyond the Philippines. Discussing the adaptability of the system to different railway infrastructures and operational conditions would enhance the paper's relevance to a broader audience.
5. Acknowledging the limitations of the current study and outlining potential areas for future research would provide a more balanced view. This could include discussing the challenges of real-world implementation, the need for further testing, or the integration of additional functionalities.
6. Given the use of deep learning for passenger analysis, the paper should address ethical considerations and data privacy issues. It is important to discuss how passenger data is handled, the measures taken to protect privacy, and the implications for the deployment of such technology in public systems.
Author Response
Good day! Thank you for taking the time to review our manuscript. Below, you will find our responses and the corresponding revisions or corrections.
- The paper could benefit from a more detailed explanation of the methodology, particularly the deep learning techniques used for passenger analysis. It would be helpful to include information on the algorithms, data sources, and validation methods to strengthen the credibility of the proposed system.
We added in our manuscript the deep learning techniques used for passenger analysis, including information on the algorithms, data sources, and validation methods on PAGE#5
- Currently, physics-informed neural networks have been widely applied in many engineering fields, as evidenced by recent publications such as "Engineering Structures, 2023, 297: 117027", "Computational Mechanics, 2021, 67: 207-230.", "Reliability Engineering & System Safety, 246, 110081." These latest advancements should also be adequately highlighted in the introduction.
We added in our manuscript the physics-informed neural networks on PAGE#2. However, for this journal, we focused on the Petri Nets.
- A comparison with existing railway monitoring and signaling systems would provide a clearer picture of the advantages and potential impact of NeuRaiSya. This could include a discussion on how NeuRaiSya addresses limitations or challenges present in current systems.
We modified our manuscript to add the comparison of existing railway monitoring and signaling systems and the NeuRaiSya on PAGE#15
- The paper should address the scalability of NeuRaiSya to larger and more complex railway networks beyond the Philippines. Discussing the adaptability of the system to different railway infrastructures and operational conditions would enhance the paper's relevance to a broader audience.
We modified our manuscript to add the scalability of NeuRaiSya on PAGE#15
- Acknowledging the limitations of the current study and outlining potential areas for future research would provide a more balanced view. This could include discussing the challenges of real-world implementation, the need for further testing, or the integration of additional functionalities.
We revised our manuscript and added about the limitations of NeuRaiSya on PAGE#16
- Given the use of deep learning for passenger analysis, the paper should address ethical considerations and data privacy issues. It is important to discuss how passenger data is handled, the measures taken to protect privacy, and the implications for the deployment of such technology in public systems.
We modified our manuscript to add about the ethical considerations & data privacy on PAGE#16
Attached is our revised manuscript to address our detailed responses.
Thank you very much.

Reviewer 2 Report
Comments and Suggestions for Authors
This article discusses dynamic modeling of the Philippine Railway monitoring process using Petri nets. The introduction describes the structure of connections of the railway monitoring system to ensure safety, efficiency and reliability. The history of the development of the railway network is considered. The following describes the main properties of Petri nets and their possible applications. An analysis is given of types of networks, such as temporary Petri nets, colored networks and stochastic ones. Software for modeling Petri nets is provided. A conceptual model for constructing an information system that analyzes and predicts monitoring non-standard situations, including a sensor system, a neural network, and a database, is presented. The structure of the network is described in detail, indicating the purpose of positions and transitions. The main properties of the Petri net were studied, such as reachability from a given zero marking and a number of others. The results include a study of the reachability property for various network models.
It should be noted that the material presented has a classical content and is devoid of any significant shortcomings.
However, the issues of using networks to optimize the structure of the transport network and its parameters depending on time intervals, a larger number of labels, and so on have not been sufficiently studied.
The conclusions obtained from the study are rather superficial and need to be improved.
An article can be accepted for publication in the journal “Sensors” only after significant revision.

Author Response
Good day! Thank you very much for dedicating your time to review our manuscript. Enclosed are our detailed responses along with the specific revisions and corrections.
1) It should be noted that the material presented has a classical content and is devoid of any significant shortcomings.
We revised our manuscript to address this comment#1 on PAGE 2, PAGE 5, PAGE 15, & PAGE 16
2) However, the issues of using networks to optimize the structure of the transport network and its parameters depending on time intervals, a larger number of labels, and so on have not been sufficiently studied.
We modified our manuscript to address this comment#2 on PAGE 5. Our manuscript for this time is focused on Petri Nets Application.
3) The conclusions obtained from the study are rather superficial and need to be improved.
We revised our manuscript to address this comment#3 on PAGE 15 & PAGE 16. Our manuscript for this time is focused on Petri Nets Application.
Attached is our revised manuscript for the detailed corrections.
Thank you very much.

Round 2
Reviewer 2 Report
Comments and Suggestions for Authors
The reviewed article has been improved in its content after revision and can be published in this form in the journal Sensors. I believe that the possibilities of using Petri nets in the areas covered in the article have not been exhausted based on the results of this work and I wish further success to their authors.
Author Response
Good day!
We would like to express our heartfelt gratitude for your time, effort, and invaluable feedback on our manuscript. Your detailed and thoughtful reviews have helped to improve the quality and clarity of our work.
We revised our manuscript, we added discussion from Page 12 to Page 15 of each figure's model to highlight the areas covered by the Petri nets. Attached here is our revised manuscript.
Thank you very much.
